# Social Responsiveness and Psychosocial Functioning in Adults with Prader–Willi Syndrome

**DOI:** 10.3390/jcm11051433

**Published:** 2022-03-05

**Authors:** Meritxell Fernández-Lafitte, Jesus Cobo, Ramon Coronas, Isabel Parra, Joan Carles Oliva, Aida Àlvarez, Susanna Esteba-Castillo, Olga Giménez-Palop, Diego J. Palao, Assumpta Caixàs

**Affiliations:** 1Mental Health Department, Corporació Sanitària Parc Taulí—CIBERSAM, 08208 Sabadell, Barcelona, Spain; fernandez.txell@gmail.com (M.F.-L.); rcoronas@tauli.cat (R.C.); iparra@tauli.cat (I.P.); dpalao@tauli.cat (D.J.P.); 2Department of Psychiatry and Forensic Medicine, Universitat Autònoma de Barcelona, 08193 Bellaterra, Barcelona, Spain; 3Institut d’Investigació i Innovació Parc Taulí (I3PT), Centres de Recerca de Catalunya (CERCA), 08208 Sabadell, Barcelona, Spain; 4Statistics Unit, Fundació Parc Taulí I3PT, CERCA, 08208 Sabadell, Barcelona, Spain; jcoliva@tauli.cat; 5Autism Disorder and Severe Mental Illness Unit, Department of Mental Health, Mutua Terrassa University Hospital, 08221 Terrassa, Barcelona, Spain; aida.pedrero@gmail.com; 6Specialized Mental Health and Intellectual Disability Department, Institut Assistència Sanitària, Parc Hospitalari Martí i Julià, 17190 Salt, Girona, Spain; susanna.esteba@ias.cat; 7Neurodevelopment Group, Girona Biomedical Research Institute “Dr. Josep Trueta” (IDIBGI), Institut Assistència Sanitària, Parc Hospitalari Martí i Julià, 17190 Salt, Girona, Spain; 8Endocrinology and Nutrition Department, Corporació Sanitària Parc Taulí, 8208 Sabadell, Barcelona, Spain; ogimenez@tauli.cat; 9Department of Medicine, Universitat Autònoma de Barcelona, 08193 Bellaterra, Barcelona, Spain

**Keywords:** Prader–Willi syndrome, autism, autism spectrum disorders, social responsiveness, social function, functionality, function

## Abstract

Although various studies have investigated symptoms of autism spectrum disorder (ASD) in Prader–Willi syndrome (PWS), little is known about the consequences of these symptoms, especially in psychosocial function. We aimed to explore ASD symptoms in adults with PWS with special attention to psychosocial functionality. This cross-sectional study included 26 adults (15 women) with PWS who attended a reference unit for rare diseases. Participants’ primary caregivers completed the Social Responsiveness Scale (SRS), and clinicians assessed multidimensional functioning with the Personal and Social Performance Scale (PSP). Impaired social responsiveness was identified in 20 (76.9%) participants, and manifest to marked difficulties in social functioning were identified in 13 (50%). Participants with impaired social responsiveness (SRS ≥ 60) had significantly worse scores in functionality measured with the PSP (U = 12.5; *p* = 0.009) and with three of the four PSP main areas. Moreover, scores for the *Social Cognition* domain of the SRS correlated positively with the *Socially useful activities* (*p* < 0.05) and *Personal and social relationships* (*p* < 0.01) main areas of the PSP. These results suggest that difficulties in social skills should be assessed in all psychosocial evaluations of patients with PWS.

## 1. Introduction

Prader–Willi syndrome (PWS) is a genetic disorder resulting from a lack of expression of paternally derived genes in the 15q11–13 region of chromosome 15. Most cases result from deletion (65–75%) or uniparental disomy (20–30%), and a few (1–3%) result from rare imprinting defects [1]. These genotypes give rise to a complex behavioral and developmental phenotype characterized by hypotonia and difficulties in feeding during infancy, followed by hyperphagia, insatiable hunger, morbid obesity, and short stature during development [2]. The estimated prevalence of PWS is between 1 in 10,000 and 1 in 30,000 live births [1]. People with PWS typically manifest compulsivity, rigidity, irritability, and social dysfunction [1,3]. General intellectual functioning in PWS ranges from average abilities to moderate intellectual disability (ID); most individuals have mild ID (IQ 55–70) [4].

The prevalence of psychopathology in individuals with ID is about four to five times higher than in the general population [5]. The most common psychiatric comorbidities in PWS are affective disorders, psychosis, obsessive-compulsive disorder, and autism spectrum disorder (ASD) [6,7]. Few studies have addressed the potential correlates of psychopathology in PWS, especially regarding functionality. The manifestations of psychopathology can vary widely. Healthcare professionals who deal with people with PWS need to be familiar with the clinical signs to identify psychological disorders.

Various studies analyzed the prevalence and relevance of symptoms of ASD in people with PWS [2,8,9,10]. In a study of 146 children with PWS, Dyckens et al. [10] reported that 18 (12.3%) were diagnosed with ASD, and 14 of these had uniparental disomy. Among adults with PWS, those diagnosed with ASD have lower IQ scores, worse social and verbal abilities, more stereotyped behaviors, and more restricted interests [10]. ASD has specific neurobehavioral manifestations, and the core symptoms of ASD include problems in the theory of mind and in social responsiveness [11].

In neurodevelopmental syndromes, social cognitive deficits often underlie difficulties in social interactions and increase the risk of worse functioning in all areas. However, little is known regarding the specific social cognitive deficits in different syndromes [12].

This study aimed to explore social responsiveness in adults with PWS and the relationship between deficits in social responsiveness and overall social functioning. We hypothesized that people with PWS would have high rates of ASD symptomatology and that symptoms of ASD would be associated with worse psychosocial functioning.

## 2. Materials and Methods

### 2.1. Design and Sample

This cross-sectional study included all adult patients with genetically diagnosed PWS attended at the endocrinology department in our reference center for rare diseases in Spain.

### 2.2. Procedure

All subjects and their legal tutors voluntarily agreed to participate and provided written informed consent. None of the patients or tutors refused to participate. All data were anonymized to preserve confidentiality.

In order to collect demographic and clinical data, including current treatments, we administered a questionnaire and measured anthropometric parameters.

### 2.3. Assessment

ID and other psychiatric disorders were diagnosed according to the fifth edition of the Diagnostic and Statistical Manual of Mental Disorders (DSM-5) [13].

To measure autistic symptomatology and traits and the severity of associated social impairment, we used the parent version of the Social Responsiveness Scale (SRS) [14]. The SRS is most often used with children and adolescents 4 to 18 years old [14], and its cross-cultural validity has been demonstrated [15]. The Spanish version of the SRS was established in 2008 after a cycle of translations and back-translations [16]. The scale comprises 65 items scored on a Likert scale ranging from 1 (not true) to 4 (almost always true), with 17 items being reverse-scored. Higher SRS scores represent more ASD-related behaviors. The SRS covers five domains (social awareness, social cognition, social communication, social motivation, and autistic mannerisms) that can be useful in clinical settings or for developing treatment plans. The domain *Social awareness* measures the ability to perceive social cues (e.g., “Is aware of what others are thinking or feeling”). The domain *Social cognition* measures the ability to interpret social cues once they are perceived (e.g., “Doesn’t recognize when others are trying to take advantage of him or her”). The domain *Social communication* measures expressive social communication (e.g., “Avoids eye contact or has unusual eye contact”). The domain *Social motivation* measures the extent to which a respondent is generally motivated to engage in social-interpersonal behavior, including elements of social anxiety, inhibition, and empathic orientation (e.g., “Would rather be alone than with others”). Finally, the domain *Autistic mannerisms* measures stereotypical behaviors or highly restricted interests (e.g., “Has an unusually narrow range of interests”). The SRS’s ease of administration and strong psychometric properties favored its widespread use in research [17]. SRS T-scores ≥ 60 indicate mild-to-moderate risk for ASD; this cutoff yields a 96.8% likelihood of a later clinical diagnosis of ASD [18].

In order to evaluate patients’ social functionality, we used the Spanish version of the Personal and Social Performance scale (PSP) [19,20]. This clinician-rated instrument evaluates patients’ social functioning in four main areas of social and individual performance (*Socially useful activities*, including work and study; *Personal and social relationships*; *Self-care*; and *Disturbing and aggressive behaviors*), independently of symptomatology. The recommended time interval for the evaluation is the last month, as in our study. Scores range from 1 to 100 and are divided into 10 equal intervals to rate patients’ degree of difficulty in functioning. Higher scores in total PSP represent better global personal and social functioning. We classified patients’ degree of difficulties in each main area in the following ranges: absent; mild; manifest, but not marked; marked; severe; or very severe (see the Appendix A). Higher scores in the main areas indicate more severe difficulties.

### 2.4. Statistical Analysis

We summarized all variables with descriptive statistics (counts and percentages, and medians and ranges, when appropriate). To determine associations between variables, we used Spearman’s rank correlation coefficient or the intraclass correlation coefficient as appropriate. To compare variables between groups, we used non-parametric statistics (Mann–Whitney U or chi-square tests, as appropriate). Statistical significance was set at *p* < 0.05. To determine the internal consistency for the SRS in our sample, we used Cronbach’s alpha. We used IBM SPSS Statistics for Windows version 21 (IBM Corp., Armonk, NY, USA) for all analyses.

## 3. Results

### 3.1. Sociodemographic and Clinical Characteristics

A total of 26 Caucasian patients (15 women; median age, 27.67 y; age range, 18.6–46.3 y) were included. All had ID according to DSM-5 criteria, classified as mild in 16 and moderate or severe in 10. Only two patients were institutionalized; the remaining 24 lived with first-degree relatives. Nearly two-thirds of the participants (*n* = 17) were employed. PWS was attributed to a deletion in 17 (65.4%) participants, to uniparental disomy in 6 (23.1%), and to imprinting defects in 3 (11.5%). The median weight was 87.8 kg (range 46.0–128.0), median height was 157 cm (range 140–190), and median body mass index (BMI) was 35.7 kg/m^2^ (range 20.4–63.4) (Table 1).

All patients were receiving different complex medical treatments, usually including psychopharmacological drugs (Appendix A).

A total of 14 (53.8%) patients received recombinant human growth hormone treatment during childhood (median duration, 75 months, range 7–168 months. None of the patients had yet received recombinant human growth hormone treatment in adulthood.

### 3.2. The Personal and Social Performance Scale

PSP scores were available for 24 cases. Overall functioning was classified as excellent in 1 (4.2%) participant, mild difficulties in 8 (33.3%), manifest to marked difficulty in 12 (50%), and severe difficulty in 3 (12.5%). Table 2 reports the levels of functionality in each main area of the PSP.

Total PSP scores were not significantly associated with sex (*p* = 0.837), age (*p* = 0.677), weight (*p* = 0.903), height (*p* = 0.753), BMI (*p* = 0.902), genetic subtype (*p* = 0.558), or the severity of ID (*p* = 0.788).

### 3.3. Social Responsiveness Scale

SRS scores indicated impaired social responsiveness in 20 (76.9%) subjects; impairment was substantial in 5 (19.2%) and severe in 15 (57.7%). By comparing patients with impaired social responsive (SRS ≥ 60) versus those without social impairment (SRS < 60) found no differences in relation to age, sex, use of recombinant human growth hormone treatment during childhood, employment situation, weight, height, or BMI, but revealed significant differences in relation to genetic subtype (*p* = 0.006), where a greater proportion of those with uniparental disomy had impaired social responsiveness, and to the severity of ID (*p* < 0.001), where a greater proportion of those with moderate or severe ID had impaired social responsiveness (Table 1).

The internal consistency of the SRS scale in our sample was good (a = 0.898).

### 3.4. Relationship between the Personal and Social Performance Scale and Social Responsiveness Scale

Participants with impaired social responsiveness (SRS scores ≥ 60) had significantly worse total PSP scores and scores in the main areas of *Self-care*, *Socially useful activities*, and *Personal and social relationships*, but not in the main area of *Disturbing and aggressive behaviors* (Table 3).

Total SRS scores tended to correlate negatively with total PSP scores, but this correlation did not reach statistical significance (r = −0.401, *p* = 0.052) (Table 3). The *Social Cognition* domain of the SRS correlated positively with two main PSP areas: *Social Activities* (r = 0.468, *p* = 0.02) and *Personal Relationships* (r = 0.516, *p* = 0.01) (Table 4).

The correlation between higher scores in the *Social Cognition* domain of the SRS with worse total functioning as measured by the total PSP score also approached statistical significance (r = −0.398, *p* = 0.054). Likewise, the positive correlation between the *Social Awareness* domain of the SRS and the *Self-care* domain of the PSP was nearly significant (r = 0.396, *p* = 0.055) (Table 3). However, in a final model for the *total PSP* scores, the SRS *Social Cognition* domain explained 16% of the total scores (R^2^ = 0.161).

## 4. Discussion

We hypothesized that people with PWS would have high rates of ASD symptomatology and that these symptoms would have negative effects on psychosocial functioning. We found that over three-quarters (76.9%) of the participants had substantially or severely impaired social responsiveness according to the SRS, and half had manifested to marked social difficulties in social functioning according to the PSP. Symptoms of ASD reflected in worse SRS scores correlated significantly with worse global functionality reflected in PSP scores. Moreover, scores in the *Social Cognition* domain of the SRS correlated significantly with those on two main areas of social functioning on the PSP (*Social Activities* and *Personal Relationships*), and participants with better social cognition also performed better in the core main functionality areas of social activities and personal relationships.

Social functionality scales were devised to develop more accurate and sophisticated instruments for measuring the level of functioning in different patients with severe mental disorders. We used the PSP to measure social function because it is multidimensional, psychometrically solid, and relatively uninfluenced by patients’ specific symptoms. The SRS was designed to measure autistic symptomatology and traits, as well as the severity of associated social impairment, in children and adolescents. Nonetheless, the SRS has been increasingly used to assess social deficits in individuals with ASD across a wide range of ages [21,22,23]. Chan et al. [24] provided empirical support for the validity of the SRS in adults with ASD. For this study, we used the Spanish version of the original SRS [14,16], although a new self-report adult version of the scale, the SRS-2 [24,25], was recently developed and has been used in different studies [26]. Although a Spanish version of the SRS-2 is currently available, we did not have access to it at the time of the evaluations.

Some of the behavioral features of PWS arguably overlap with those found in ASD, a neurodevelopmental disorder characterized by impaired social communication and highly repetitive or restricted behaviors and interests [13]. A small percentage of people with ASD have alterations in chromosome 15q11.2–q13, the critical region for PWS [27]. A systematic review of studies including a total of 786 participants with PWS found that 210 (26.7%) met the criteria for ASD [2]; this prevalence is much higher than the 1.5% estimated prevalence of ASD in the general population [28].

In people with PWS, early detection of symptoms of autism might enable specific interventions to prevent the exacerbation of more specific symptoms of ASD [9] and to improve global functioning. Our findings suggest that impaired social responsiveness related to ASD symptomatology is common in adults with PWS. Individuals with impaired social cognition might benefit from more help in social interactions. It is important to attempt to understand individuals with impaired social cognition and to adjust attitudes when dealing with them to prevent frustration and maladaptive behaviors [12].

We found significant positive linear correlations between the *Social Cognition* domain of the SRS and two main areas of function in the PSP (*Socially useful activities* and *Personal and social relationships*), as well as a nearly significant negative correlation with the total PSP score, suggesting that greater difficulties in social cognition are intertwined with difficulties in social activities and relationships that would hinder the of overall ability people with PWS to function socially. The SRS *Social Cognition* domain measures the ability to interpret social cues once they have been perceived and represents aspects of cognitive interpretation of reciprocal social behavior. Impaired social cognition, including deficits in mentalizing and metacognition that affect people with ASD, might explain a large portion of the social impairment observed in people with PWS. However, the impact of the *Social Cognition* domain of the SRS alone on total functioning in our sample was relatively small, explaining only 16.1% of the total PSP score in our sample (r^2^ = 0.161).

We also found a significant positive correlation between the *Social Awareness* domain of the SRS and the *Self-care* main area of the PSP. The *Social Awareness* domain refers to the ability to perceive social cues, and this part of the SRS comprises items representing the sensory aspects of reciprocal social behavior. Our findings in this domain could be related to the deficits in insight dimensions found in our group’s previous studies in patients with PWS, who showed good awareness of their illness and its core symptoms (e.g., obesity/excess weight or excessive appetite) and of the effects of psychotropic medication, but lacked awareness of the social consequences of their illness (e.g., excessive food intake) [29,30]. This profile of insight may have relevant clinical implications, as suggested in the combined results for functionality and social responsiveness. If awareness of illness is a metacognitive ability, both good insight and better social awareness could improve self-care.

Higher total SRS scores correlated with poorer global social functionality measured with the PSP, although this correlation did not reach statistical significance (*p* = 0.052), suggesting social skills are fundamental for functionality in broad terms regardless of whether other limitations are present.

Most cases of PWS are sporadic, and although PWS is equally prevalent in males and females [31], our sample included 15 women and 11 men. ASD is more commonly diagnosed in males (3:1 male:female ratio, according to current estimates), although there seems to be a high likelihood of a gender bias in the diagnosis of ASD [32]. According to the extreme male brain theory, autistic traits in general ASD populations would represent an exaggerated version of the psychological profile typically observed in male individuals and would be associated with increased prenatal exposure to sexual steroids that impact prenatal brain development [33,34,35,36]. This theory is supported by the known role of sex hormones in brain organization and function during developmental periods, in addition to their activation role in adults [37]. In fact, gender identification with natal sex is lower in people with ASD, especially in natal females [38], and girls with ASD could camouflage symptoms or imitate neurotypical traits [39,40,41]. On the other hand, some studies in PWS have found a higher prevalence among males [10]. We consider that it is very unlikely that the impaired social responsiveness observed in our sample was affected by sex because the particularities of PWS and the impact of the severity of ID precluded participants’ camouflaging ASD traits; moreover, SRS scores were calculated from parents’ reports and did not include subjective information from participants’ themselves.

Similarly, we found no association between age and autistic traits in our sample of adults with PWS. Previous studies have suggested that autistic traits identified in children with PWS become gradually more prominent with aging [42], as reflected in significantly higher scores on the Pervasive Developmental Disorders—Autism Society Japan Rating Scale [43] in adolescents with PWS than in young adults with PWS.

Bennet et al. [2] suggested that many symptoms of PWS probably overlap with symptoms of idiopathic ASD (although they may differ qualitatively) and that, therefore, the prevalence of ASD symptomology in PWS is probably overestimated, given that the instruments used to identify ASD symptoms can lack specificity in individuals with PWS. More research is needed to determine the most appropriate tools for assessing ASD symptoms in children and adults with PWS.

We found that social responsiveness was more impaired in individuals with maternal uniparental disomy and in those with moderate or severe ID. Both these findings are in line with those reported by Dykens et al. [10] and Dimitropoulos et al. [44]. These two aspects may explain some of the functional results, especially when they occur together, although the subgroups in our sample were very small (one patient, five patients) (Table 1), increasing the risk of bias in the statistical analysis, and larger studies are needed to enable conclusions. People with PWS due to maternal uniparental disomy are thought to be at greater risk of autistic symptoms than those with deletions on the paternally inherited chromosome 15 because of the maternally inherited duplication and thus overexpression of genes in the 15q11–13 region [45]. As Dimitropoulos et al. [44] pointed out, identifying common behaviors between individuals with PWS due to maternal uniparental disomy and ASD may further indicate the importance of overexpression of the 15q11–13 regions in increasing the risk of ASD and specifically in ASD-related social impairment.

It is important to identify difficulties and limitations related to other mental health issues in patients with PWS, as they will have implications for treatment and interventions [9]. Likewise, it is important to take common comorbidities associated with PWS into account when assessing patients for symptoms of ASD. Defining the spectrum of socio-communicative deficits in PWS is crucial for developing the most accurate intervention strategies. If social deficits in PWS are similar to those in ASD, evidence-based interventions for ASD patients can be generalized to PWS patients [2]. Given the prevalence of social difficulties in adults with PWS, it would be prudent for professionals to assess social deficits in patients with PWS from an early age. Early detection, even in young adults, can enable specific interventions to improve social functioning in people with PWS.

Finally, we found no relationship between BMI and SRS or PSP scores. These findings are in line with those reported by Dykens et al. [10], who found no association between BMI and the diagnosis of ASD. As hyperphagia, insatiable hunger, and morbid obesity are among the core symptoms of PWS, we could hypothesize that social responsiveness could be related to some of the neurobehavioral symptoms of PWS (e.g., restricted or repetitive behaviors and interests, challenging behaviors such as tantrums or self-injurious behaviors, and especially impaired social communication), but not with others (e.g., hyperphagia, insatiable hunger, or morbid obesity). Along these lines, in a study in adults with eating disorders (44 acute Anorexia Nervosa and 49 recovered Anorexia Nervosa), Kerr-Gaffney et al. [26] also found that SRS-2 scores were positively associated with eating-disorders-associated psychopathology and functional impairment, but not with BMI or illness duration. Future studies could explore the influence of ASD symptoms on specific food behaviors and BMI in people with PWS.

## 5. Strengths and Limitations

Unlike most previous studies of ASD in PWS, the current study focused on ASD symptoms in adults, thus providing much-needed data about a relatively unknown topic. Nevertheless, various limitations of our study must be taken into consideration. Our small sample makes subgroup analyses difficult, and the cross-sectional design only allows conclusions about associations. The PSP has certain drawbacks in PWS. In the total PSP score, the *Disturbing and aggressive behaviors* area has more weight than the other three main areas; however, our participants’ scores in this area were not especially high. Moreover, the SRS was designed for use in children and adolescents, and the samples used to validate this scale in adults have comprised mainly non-genetically diagnosed individuals; thus, caution is warranted in extending its validity to other populations. Finally, it is difficult to compare our results with those of previous studies due to differences in the instruments used to measure social responsiveness and social functionality.

## 6. Conclusions

Impaired social responsiveness related to ASD symptomatology is prevalent in adults with PWS. People with PWS scoring ≥ 60 on the SRS had significantly worse scores on the PSP, suggesting that symptoms of ASD are related to worse social functioning. Moreover, scores in the *Social Awareness* domain of the SRS correlated with scores in the *Self-care* area of the PSP, suggesting that efforts to improve social awareness might help improve self-care. Our results suggest that evaluating difficulties in social skills is fundamental for the global assessment of functionality.

## Figures and Tables

**Table 1 jcm-11-01433-t001:** Sociodemographic, clinical, and psychometric variables in adults with Prader–Willi syndrome.

	Total Sample (*n* = 26)	Normal SRS Scores (*n* = 6)	SRS Scores Indicating Impairment * (*n* = 20)	Normal SRS vs. SRS Impairment U; *p*/c^2^; *p* (df)
Age, in years; median (range)	27.67	30.36	26.87	40.0; 0.242
	(18.6–46.3)	(19.5–38.7)	(18.6–46.3)	
Sex:				
– Female, *n* (%)	15 (57.7)	3 (50)	12 (60)	0.61; 0.433 (1)
– Male, *n* (%)	11 (42.3)	3 (50)	8 (40)	
Employment status, *n* (%)				7.53; 0.117 (1)
– Employed/student/housewife	17 (65.4)	5 (83.3)	12 (60.0)	
– Unemployed/pensioner	9 (34.6)	1 (16.7)	8 (40.0)	
Weight, in kg; median (range)	87.80	88.01	87.74	56.0; 0.836
	(46.0–128.0)	(46.0–128.0)	(52.9–126.1)	
Height, in cm; median (range)	157	151	158	41.5; 0. 268
	(140–190)	(140–160)	(140–190)	
BMI, in kg/m^2^; median (range)	35.68	37.67	35.08	46.0; 0.421
	(20.4–63.4)	(20.4–49.7)	(24.1–63.4)	
BMI according to WHO classification, *n* (%)				7.35; 0.236 (4)
– Normal weight	3 (11.5)	1 (16.7)	2 (10)	
– Overweight	3 (11.5)	–	3 (15)	
– Obesity class I	9 (34.6)	1 (16.7)	8 (40)	
– Obesity class II	4 (15.4)	1 (16.7)	3 (15)	
– Obesity class III	7 (26.9)	3 (50)	4 (20)	
Genetic subtype, *n* (%)				**12.53; 0.006 (2)**
– Paternal microdeletions	17 (65.4)	4 (66.7)	13 (65)	
– Uniparental maternal disomy	6 (23.1)	1 (16.7)	5 (25)	
– Imprinting defects	3 (11.5)	1 (16.7)	2 (10)	
ID (DSM-5), *n* (%)				**11.61; <0.001 (2)**
– Mild ID	16 (61.5)	5 (83.4)	11 (55)	
– Moderate/severe ID	10 (38.5)	1 (16.6)	9 (45)	

SRS: Social Responsiveness Scale. * SRS scores ≥ 60. U: Man–Whitney U. χ^2^: chi-square test. *p*: significance. df: degrees of freedom. DSM-5: Diagnostic and Statistical Manual of Mental Disorders (APA, 2014). ID: intellectual disability. VAS: visual analog scale. BMI: body mass index. WHO: World Health Organization. WHO Classification of BMI: Normal (18.5–24.99), Overweight (25–29.99), Obesity Class I (30–34.99), Obesity Class II (35–39.99), Obesity Class III (≥40).

**Table 2 jcm-11-01433-t002:** Score for main areas of functionality in the Personal and Social Performance scale in 24 adults with Prader–Willi syndrome.

Level of Difficulty in Main Areas of the PSP *Difficulties*	*Absent*	*Mild*	*Manifest, but Not Marked*	*Marked*	*Severe*	*Very Severe*
*Self-care*: Cases (%)	8 (33.3)	9 (37.5)	5 (20.8)	2 (8.3)	-	-
*Socially useful activities*: Cases (%)	5 (20.8)	8 (33.3)	3 (12.5)	5 (20.8)	3 (12.5)	-
*Personal and social relationships*: Cases (%)	5 (20.8)	5 (20.8)	7 (29.2)	3 (12.5)	4 (16.7)	-
*Disturbing and aggressive behaviors*: Cases (%)	11 (45.8)	10 (41.7)	2 (8.3)	1 (4.2)	-	-

PSP: The Personal and Social Performance scale.

**Table 3 jcm-11-01433-t003:** Bivariate relationships between SRS domains and PSP main areas of functionality in adults with Prader–Willi syndrome.

	Total Sample (*n* = 24)	Normal SRS Scores (*n* = 5)	SRS Scores Indicating Impairment * (*n* = 19)	Normal SRS vs. SRS Impairment U; *p*
SRS *Social Awareness*, median (range)	8.54	3.50	10.05	**3.5; <0.001**
	(1–14)	(1–7)	(3–14)	
SRS *Social Cognition*, median (range)	15.46	8.16	17.65	**2.5; <0.001**
	(5–24)	(5–13)	(11–24)	
SRS *Social Communication*, median (range)	25.27	10.83	29.6	**3.5; <0.001**
	(4–42)	(4–21)	(18–42)	
SRS *Social Motivation*, median (range)	11.50	5.66	13.25	**5.0; <0.001**
	(1–23)	(1–7)	(4–23)	
SRS *Autistic Mannerism*, median (range)	17.23	9.33	19.6	**7.0; <0.001**
	(4–32)	(4–13)	(8–32)	
SRS Total scores, median (range)	77.23	37.66	89.10	**0.0; <0.001**
	(28–118)	(28–58)	(66–118)	
PSP *Self-care*, median (range)	1.04	0.20	1.26	**16.0; 0.018**
	(0–3)	(0–1)	(0–3)	
PSP *Socially useful activities*, median (range)	1.70	0.40	2.05	**13.0; 0.012**
	(0–4)	(0–1)	(0–4)	
PSP *Personal and social relationships*, median (range)	1.83	0.40	22,105	**10.0; 0.005**
	(0–4)	(0–1)	(0–4)	
PSP *Disturbing and aggressive behaviors*, median (range)	0.70	0.40	0.78	36.0; 0.446
	(0–3)	(0–1)	(0–3)	
PSP Total scores, median (range)	61.58	82.80	56.0	**12.5; 0.009**
	(25–98)	(66–98)	(25–78)	

SRS: Social Responsiveness Scale. * SRS scores ≥ 60. U: Mann–Whitney U. PSP: Personal and Social Performance scale.

**Table 4 jcm-11-01433-t004:** Spearman correlations between domains of the Social Responsiveness Scale and main areas of functionality of the Personal and Social Performance scale in 26 adults with Prader–Willi syndrome.

*Domains/Main Areas*		PSP: *Self-Care*	PSP: *Socially Useful Activities*	PSP: *Personal and Social Relationships*	PSP: *Disturbing and Aggressive Behaviors*	PSP: Total Scores
SRS *Social Awareness*	*r.*	**0.396**	0.327	0.305	0.086	−0.366
	*Sig.*	**0.055**	0.119	0.148	0.690	0.079
SRS *Social Cognition*	*r.*	0.010	**0.468** *	**0.516** **	0.077	−0.398
	*Sig.*	0.962	**0.021**	**0.010**	0.721	**0.054**
SRS: *Social Communication*	*r.*	0.133	0.268	0.262	0.183	−0.293
	*Sig.*	0.535	0.206	0.215	0.392	0.164
SRS: *Social Motivation*	*r.*	0.379	0.137	0.168	0.141	−0.232
	*Sig.*	0.068	0.522	0.432	0.512	0.276
SRS: *Autistic Mannerism*	*r.*	0.173	0.296	0.238	0.269	−0.335
	*Sig.*	0.418	0.160	0.262	0.204	0.110
SRS: Total Scores	*r.*	0.243	0.336	0.323	0.247	−0.401
	*Sig.*	0.253	0.108	0.123	0.245	**0.052**

SRS: Social Responsiveness Scale. PSP: The Personal and Social Performance scale. r.: Correlation coefficient. Sig.: Signification. In bold are correlations of interest. Significance: * 0.05, ** 0.01.

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
