# Peer review of "Social Responsiveness and Psychosocial Functioning in Adults with Prader–Willi Syndrome"

_jcm, 2022, doi:10.3390/jcm11051433_

Round 1
Reviewer 1 Report
Thank you for the opportunity to review the manuscript “Social responsiveness and psychosocial functioning in adults with Prader-Willi syndrome” by Meritxell Fernández-Lafitte et al. It is an interesting and novel analysis of influence of autism spectrum disorder symptoms in adults with PWS on their psychosocial functionality.
However, there are some issues that need reconsideration to make the paper more precise.
- Abstract: “PWS adults”- please change to “adults with PWS”.
- Please add a paragraph regarding the studies, clinical and experimental, that try to explain the background of ASD in patients with PWS (hypothalamic disorders, genetic background).
- There is lack of information regarding recombinant human growth hormone treatment in the study population and the possible correlation with the results.
- Lack of information about any other treatment, including psychiatric therapy.
- Patients with UPD are very small groups (1 pt, 5 pts), I would recommend adding a commentary about this issue and possible bias in the statistical analysis.
- Paragraph 3. Social Responsiveness Scale- p values do not match with p values in the Table 1.
- If the Authors conclude that UPD create a higher risk for psychosocial disturbances please discuss this hypothesis further.
- Lack of the final paragraphs regarding- informed consent, ethical committee etc.
Author Response
Dear Reviewer,
Thank you for all your comments and suggestions. We have incorporated all of them in the revised manuscript, and we are sure that they have helped us improve our manuscript substantially.
Changes to the manuscript are highlighted in yellow.
1. Abstract: “PWS adults”- please change to “adults with PWS”.
This term has been changed throughout the manuscript.
2. Please add a paragraph regarding the studies, clinical and experimental, that try to explain the background of ASD in patients with PWS (hypothalamic disorders, genetic background).
Thank you for this suggestion. This is a fascinating topic, but we consider it might be beyond the scope of the current study. Although various hypotheses have been put forth to explain the hypothalamic and genetic background of ASD in patients with PWS [6] [8] [27], most of these involve an association between the genetic risk for autism and regions characteristic for PWS (15q11–13) [45].
3. There is lack of information regarding recombinant human growth hormone treatment in the study population and the possible correlation with the results.
We have collected the data about recombinant human growth hormone treatment for each patient, and we have included the data in the Results section.
4. Lack of information about any other treatment, including psychiatric therapy.
Among the most common medical treatments administered were oral antidiabetics (n=6), insulin (n=3), antihypertensives (n=4), lipid-lowering agents (n=2), levothyroxine (n=3), and other hormonal treatments (n=4) in different combinations. Among the most common psychopharmacological treatments administered were different antipsychotic medications (n=7; median chlorpromazine equivalent 100 mg/d, range 25‒280 mg/d), antidepressants (n=14; fluoxetine in 4, sertraline in 4, other antidepressants in 6), benzodiazepines (n=3, median diazepam equivalents 2.5 mg/d, range 2.5‒5 mg/d), topiramate (n=10, median dosage 250 mg/d, range 50‒500 mg/d), and zonisamide (n=3, median dosage 50 mg/d, range 50‒350 mg/d) .
We consider that including all this information in the results section would distract readers rather than help clarify our message. Thus, we have included only the following information:
All patients were receiving different complex medical treatments, usually including psychopharmacological drugs.).
A total of 14 (53.8%) patients received recombinant human growth hormone treatment during childhood (median duration, 75 months, range 7‒168 months. None of the patients had yet received recombinant human growth hormone treatment in adulthood.
If you consider it very important, we could include a chart detailing the treatments for each individual patient in the supplementary material.
5. Patients with UPD are very small groups (1 pt, 5 pts), I would recommend adding a commentary about this issue and possible bias in the statistical analysis.
We have included this information in the discussion and again in the limitations section.
6. Paragraph 3. Social Responsiveness Scale- p values do not match with p values in the Table 1.
Thanks for pointing this out. Sorry about the confusion. We have corrected this mistake.
7. If the Authors conclude that UPD create a higher risk for psychosocial disturbances please discuss this hypothesis further.
We discuss this issue in relationship with all the possible disturbances related to UPD.
8. Lack of the final paragraphs regarding- informed consent, ethical committee etc.
We apologize for this omission. We have included all the required information in the revised manuscript.
Thank you for your contributions. Please note that the manuscript has been revised by an editor from the United States who has extensive experience with biomedical texts.
The Authors
Reviewer 2 Report
This manuscript is well-written and describes an interesting study that expands understanding of PWS by examining social responsiveness and psychosocial functioning in adults.
I have one specific comment. Table 1 includes a comparison of BMIs between adults with normal SRS scores and scores indicating impairment. The difference in BMIs was not significant. The discussion goes on to include mention of social responsiveness, particularly social awareness and self care, including eating behavior. I would like to see more discussion of the lack of difference in BMI values in adults between participants with normal SRS scores and scores indicating impairment and what your findings might mean in relation to therapies to improve social responsiveness and likely impacts of eating behavior and obesity in adults with PWS.
Author Response
Dear Reviewer,
Thank you for all your comments and suggestions; we are sure that they have helped us improve our manuscript substantially. Changes to the manuscript are highlighted in yellow.
Comment: “Table 1 includes a comparison of BMIs between adults with normal SRS scores and scores indicating impairment. The difference in BMIs was not significant. The discussion goes on to include mention of social responsiveness, particularly social awareness and self care, including eating behavior. I would like to see more discussion of the lack of difference in BMI values in adults between participants with normal SRS scores and scores indicating impairment and what your findings might mean in relation to therapies to improve social responsiveness and likely impacts of eating behavior and obesity in adults with PWS”.
Answer: We discuss these aspects in the Discussion of the revised manuscript.
Thank you again for your valuable input.
The Authors
Round 2
Reviewer 1 Report
Dear Authors,
Thank you for accepting the suggestions for improvement of the manuscript.
Please clarify:
- Results
“All patients were receiving different complex medical treatments, usually including psychopharmacological drugs.).”- how many patients received the psychiatric treatment? I would indeed recommend adding the supplementary material regarding the treatment.
- Discussion
“Along these lines, in a study in adults with eating disorders, Kerr-Gaffney et al [26]”- please explain what kind of eating disorders (anorexia nervosa?).
Author Response
Dear Reviewer:
Thank you for all your news comments and suggestions. We have incorporated both of them in the revised manuscript. Changes to the new manuscript draft are also highlighted in yellow.
First: “Please clarify: Results “All patients were receiving different complex medical treatments, usually including psychopharmacological drugs.).”- how many patients received the psychiatric treatment? I would indeed recommend adding the supplementary material regarding the treatment”.
We include the information in Supplementary Material.
Second: “Discussion: “Along these lines, in a study in adults with eating disorders, Kerr-Gaffney et al [26]”- please explain what kind of eating disorders (anorexia nervosa?).”
Yes ! We included the characteristics of the sample in Kerr-Gaffney et al [26].
Thank you,
The Authors